# DHODH Inhibition Suppresses *MYC* and Inhibits the Growth of Medulloblastoma in a Novel In Vivo Zebrafish Model

**DOI:** 10.3390/cancers16244162

**Published:** 2024-12-13

**Authors:** Ioanna Tsea, Thale Kristin Olsen, Panagiotis Alkinoos Polychronopoulos, Conny Tümmler, David B. Sykes, Ninib Baryawno, Cecilia Dyberg

**Affiliations:** 1Division of Pediatric Oncology and Pediatric Surgery, Department of Women’s and Children’s Health, Karolinska Institutet, 171 77 Stockholm, Sweden; 2Department of Immunology, Genetics, and Pathology, Uppsala University, 753 10 Uppsala, Sweden; 3Center for Regenerative Medicine, Massachusetts General Hospital, Boston, MA 02114, USA

**Keywords:** paediatric cancers, brequinar, DHODH, relapse, medulloblastoma, targeted therapy

## Abstract

This study explores the use of brequinar (BRQ), a drug that blocks the activity of dihydroorotate dehydrogenase (DHODH), an enzyme involved in producing the building blocks needed for DNA and RNA in cells. In aggressive group 3 medulloblastoma, a childhood brain tumour, DHODH activity is linked to the MYC gene, which drives tumour growth. Our results demonstrate that by inhibiting DHODH with brequinar, tumour cell growth and *MYC* levels are significantly reduced. This effect was observed in both laboratory cell cultures and in a novel zebrafish model, which mimics the tumour’s behaviour in a living organism. This approach could offer a new, more targeted treatment for high-risk medulloblastoma, and provide a rapid and effective model for testing other potential therapies.

## 1. Introduction

Medulloblastoma (MB) is the most common paediatric brain malignancy in children younger than 15 years of age. MB comprises four molecular subtypes, including Sonic Hedgehog (shh), Wingless (wnt), group 3, and group 4, all characterised by different genetic, epigenetic, and genomic alterations, prognoses, and treatment approaches [1]. Of these four subtypes, group 3 tumours remain the most poorly understood, and are associated with the worst prognosis, displaying high rates of recurrence and a propensity to metastatic spread [2].

Group 3 comprises 25% of all MBs. So far, there has not been a commonly overexpressed pathway identified in group 3, although isochromosome 17q is often observed. In addition, several genetic alterations, including *MYC* amplification, *OTX2* amplification, *SMARCA4* and *GFL1B* mutation, and *GLI1* enhancer activation, have been observed. The incidence of metastasis at diagnosis is 40–45% in group 3 MB [3].

Unfortunately, around 30% of MB patients (mostly belonging to the shh, group 3, or group 4 subgroup) will relapse, which is almost always incurable. However, the molecular signalling pathways involved in MB metastasis are still largely unknown [4].

Efforts to characterise the biology and pathogenesis of group 3 MBs point to the importance of the *MYC* oncogene, as it is frequently overexpressed or amplified [5]. MYC is a transcription factor whose target genes are involved in, among others, apoptosis, metabolism, and cell growth [6]. Despite the recognition of *MYC* as a driver of group 3 MB, attempts at therapeutically targeting MYC have largely been unsuccessful. Newer strategies have focused on suppressing MYC function through the prevention of its binding to partner proteins or the inhibition of associated pathways [7,8]. One pathway upregulated by MYC in the context of tumour growth and proliferation is that of pyrimidine synthesis [9]. Dihydroorotate dehydrogenase (DHODH) catalyses the conversion of dihydroorotate to orotate during de novo pyrimidine synthesis [10]. DHODH catalyses the only step in the pyrimidine synthesis pathway that occurs inside the mitochondria [9,10]. To meet the demands of sustained proliferation and growth, tumour cells are dependent on pyrimidine biosynthesis to provide building blocks for DNA and RNA, and to contribute to carbohydrate and lipid metabolism [11].

Due to MYC-mediated metabolic reprogramming, tumour cells become dependent on the de novo biosynthesis pathway to sustain cell proliferation, suggesting that these cells are highly sensitive to inhibitors that target this metabolic pathway [12]. To this end, clinical trials investigating the efficacy of DHODH inhibition have been reported for AML (ClinicalTrials.gov: NCT03404726), advanced lymphoma (ClinicalTrials.gov: NCT03834584), and recurrent glioma (ClinicalTrials.gov: NCT05061251).

Here, we investigate BRQ, a potent and specific DHODH inhibitor, as a potential therapy to target group 3 MB. Even as a monotherapy, BRQ can induce apoptosis of MB cell lines in the nanomolar range, as well as reduce *MYC* expression in group 3 MB cells. To assess the potential of BRQ in vivo to evaluate the toxicity of the compound, we established a zebrafish xenograft assay of group 3 MB, which we successfully used for rapid pharmacological testing. BRQ has a potent apoptotic effect in vivo at non-toxic concentrations for zebrafish embryos and demonstrates potential as a treatment candidate for group 3 MB.

## 2. Materials and Methods

### 2.1. Survival and Gene Expression Data

Survival and gene expression data were collected from the publicly available R2 datasets of Cavalli [13]; Williamson [14], for human MB data; and the Cancer Cell CCLE DepMap 21q4 dataset [15], for MB and other paediatric cancer cell line data. Survival plots were obtained using the Kaplan–Meier feature of the R2 database [16], using the median value as the cut-off mode, with the minimum group size set to 8. Overall survival was defined as days from diagnosis until death from any cause, or end of follow-up.

### 2.2. Cell Culture and Reagents

Seven MB cell lines, derived from different subgroups (shh, group 3, and group 4), with different genetic characteristics, were used in the study. All cell lines, along with growth patterns, group classification, and other characteristics, are listed in Appendix A. The cells were purchased from ATCC (ATCC-LGC Standards, Middlesex, UK), except for D425, D458, Med8a, and UW228-3, which were kindly provided by Dr. M. Nistér (Karolinska Institutet, Sweden), as well as MB-LU-181, which was previously established by our group [16]. The cells were grown and maintained as follows: Med8a in Dulbecco’s modified Eagle’s medium (DMEM); DAOY in Minimum Essential Media (MEM); D425 and D458 in Richter’s improved MEM with zinc/DMEM; UW228-3 in DMEM/F12, supplemented with 10% (or 15% for Med8a, D425, D458) heat-inactivated foetal bovine serum (FBS); and 2 mM L-glutamine, 100 IU/mL penicillin G, and 100 μg/mL streptomycin (all from Life Technologies Inc., Thermo Fisher Scientific, Stockholm, Sweden). CHLA-01-MED and CHLA-01R-MED spheres were cultured in DMEM/F12 with 20 ng/mL human recombinant epidermal growth factor (EGF, Chemicon, Merck Millipore, Solna, Sweden), 20 ng/mL human recombinant basic fibroblast growth factor (bFGF) (Gibco Life Technologies Inc.), and B-27 Supplement (Gibco, Life Technologies Inc., Stockholm, Sweden), in addition to L-glutamine, streptomycin, and penicillin, supplemented as described above. MB-LU-181 spheres were grown in Ultra-Low attachment 6-well plates (Corning, VWR, Spånga, Sweden), in UltraCULTURE™ cell culture medium (Lonza BioWhittaker Inc., VWR, Basel, Switzerland), supplemented with 20 ng/mL EGF, plus L-glutamine, streptomycin, and penicillin, as described above. All cells were grown at 37 °C in a humidified 5% CO_2_ atmosphere. All media were purchased from Gibco (Life Technologies Inc.). The identities of the cell lines were verified by short tandem repeat genetic profiling using the AmpFlSTR^®^ IdentifilerTM PCR Amplification Kit 2015 (Applied Biosystems, Thermo Fisher Scientific, Stockholm, Sweden), and they were routinely tested for mycoplasma (Mycoplasmacheck, Eurofins Genomics, Ebersberg, Germany). All experiments were executed in Opti-MEM, supplemented with glutamine, streptomycin, and penicillin, except for MB-LU-181 cells, which were seeded in their standard growth medium. For more detailed information see [17].

BRQ was a kind gift from Dr. Sykes.

### 2.3. Cell Viability Analysis

The effects of DHODH inhibition on cell viability were evaluated using a colourmetric formazan-based cell viability assay (WST-1; Roche, Sigma-Aldrich Stockholm, Sweden). Cells were seeded in Opti-MEM onto 96-well plates, with 5000–10,000 cells per well, and treated for 72 h. Absorbance was measured at 450 and 650 nM using a VersaMax reader (Molecular Devices, San Jose, CA, USA). Cell viability is presented as the % of untreated control cells. All concentrations were at least tested in triplicate.

### 2.4. RNA Sequencing, Differential Gene Expression Analysis, and Gene Set Enrichment Analysis

The MB cell lines D425 and D458 were seeded in 6-well plates (300,000 cells/well), left to attach, and treated with BRQ 4.3 nM (D425), 15.4 nM (D458) (IC50s from the viability assays at 72 h), or vehicle (0.01% DMSO), for 24 h or 72 h, in triplicate. RNA was purified using the RNeasy kit (Qiagen, Sollentuna, Sweden). The total RNA was subjected to quality control with Agilent Tapestation, according to the manufacturer’s instructions. Sequencing libraries were constructed using the Truseq stranded mRNA sample preparation protocol (including mRNA isolation, cDNA synthesis, ligation of adapters, and amplification of indexed libraries). The yield and quality of the amplified libraries were analysed using qubit by Thermo Fisher Scientific and the Agilent Tapestation. Indexed cDNA libraries were normalised and combined, and the pools were sequenced on the Illumina Nextseq 2000 platform (Illumina, CA, USA) for a 100-cycle v3 sequencing run, generating 75 bp single-end reads. Base calling and demultiplexing were performed using CASAVA software (version 1.5) with default settings, generating Fastq files for further downstream mapping and analysis. Raw sequencing Fastq files were aligned to the GRCh38 human reference genome using the STAR 2 pass approach [18]. Aligned reads were quantified using htseq-count [19], and differential gene expression was performed using DEseq2 [20]. After DEseq2 analysis, genes were ranked by adjusted *p*-value, and the sign of the log fold-change and gene set enrichment analysis was performed using GSEA software (version 4.3.3) [21]; the hallmark gene sets are from MSigDB.

### 2.5. In Vivo Zebrafish Xenografts

Zebrafish were housed in self-cleaning 3.5 L tanks at a density of five fish per litre, in a centralised recirculatory aquatic system (Tecniplast, Buguggiate, Italy). Basic water parameters were continuously surveyed, and automatically adjusted to a temperature of 28 °C, a conductivity of 1200 µS/cm, and a pH of 7.5. A lighting scheme of 14-h light/10-h dark with 20-min dawn and dusk periods was used.

D425 and D458 cells were transduced with a lentiviral construct that was subcloned into a lentivector (HIV-1-derived plasmid) containing a GFP puromycin marker (GFP-Puro) (AMSBIO, Alkmaar, The Netherlands), and selected under puromycin treatment, following FACS sorting, to obtain GFP-expressing cells, which were cultured as described above. Prior to injection into the zebrafish embryos, cells were detached and passed over a 40 μm sterile filter to acquire a single-cell suspension. Zebrafish embryos were raised in E3 media-supplemented water containing 30 mg/L of phenylthiourea (PTU) to block pigmentation. Before transplantation, cells were pelleted (2000 rpm, 90 s) and resuspended in a minimal volume of buffer. A microcapillary (TW100-4, World Precision Instruments, Frieberg, Germany) was pulled using a Sutter P1000 needle puller, and the cell suspension was loaded into the microcapillary connected to a Femtojet^®^ 4× (Eppendorf, Hamburg, Germany). A 96-well plate (Ibidi, Gräfelfing, Germany) was prepared with 250 µL of 1% agarose in 1x E3 medium per well, with a mould placed on top of the plate until the agarose had solidified [22,23]. Each 4 hpf embryo was injected with approximately 100–300 cells in the centre of the cell mass according to van Bree et al., 2024 [23]. After transplantation, the embryos were collected in E3 medium and raised at 33 °C. Embryos with intracranial tumours were selected for treatment 24 h after transplantation. BRQ was added directly into the E3 medium at a final concentration of 2.5 µM. DMSO was used as a vehicle control. Single embryos were distributed with 150 µL of exposure medium (160 µg/mL tricaine, 30 mg/L phenylthiourea in E3 medium) into the wells, manually oriented into position, and imaged at 24 h and 72 h of treatment using ImageXpress Nano (Molecular Devices, Jan Jose, CA, USA) at 10×. After the imaging at 72 h, the plate was moved into a wet-chamber and placed inside of an incubator at 33 °C. At 120 hpf, zebrafish embryos were anaesthetised with 2.4 mM Tricaine and fixed in 4% formaldehyde.

Drug tolerance was measured by the exposure of 24 h zebrafish embryos to E3 medium containing increasing concentrations of BRQ, including one group exposed to DMSO and one control group with only E3 medium. Embryos were monitored every 24 h for a total of 72 h of exposure to deduce signs of toxicity caused by the treatments. The level of toxicity was noted on a numerical scale, where 0 = no toxicity, 1 = toxicity noted only in one or two fish, 2 = weak signs of toxicity in several fish but no deaths, 3 = strong signs of toxicity but no deaths, and 4–5 = lethal doses of drugs, with 5 suggesting that the effect might have been instant.

To assess tumour growth in the zebrafish xenografts, we developed a custom image-based script for quantification of the tumour areas in the acquired images using the analyse particles tool in ImageJ2 (2.14.0/1.54f). Images were automatically analysed using the macro with a threshold for signal set at 4× the average background in the plate; the settings were otherwise kept constant during the analysis, and the analysis was performed blind. For more detailed information, see [23].

### 2.6. Immunohistochemistry (IHC) of Zebrafish Tumour Sections and Cell Lines

D425 and D458 cell lines, treated with BRQ at concentrations equal to their IC50, were collected after 24 and 72 h of treatment, pelleted, and fixed for 12–16 h in 4% PFA. After paraffin embedding, cell pellets were processed for immunohistochemical staining using the following primary antibodies: Rabbit Ki-67 (Thermo Fisher Scientific, MA5-14520, 1:400, Stockholm, Sweden), Rabbit-cleaved b-3 (Cell Signaling Technology, #9664, 1:2000, Leiden, The Netherlands), and Mouse-cleaved PARP (Cell Signaling Technology, #32563S), in concentrations of 1:400, 1:500, and 1:200, respectively) Additionally, Zebrafish xenograft samples were fixed using 4% PFA overnight, paraffin-embedded, and processed for IHC staining with the following primary antibodies: Rabbit Ki-67 (Thermo Fisher Scientific, MA5-14520, 1:400, Stockholm, Sweden) and Rabbit Caspase-3 (Cell Signaling Technology, #9662, Leiden, The Netherlands), in concentrations of 1:400 and 1:500, respectively.

Paraffin blocks were sectioned for IHC in 4-μm wide sections and were deparaffinised using xylene and rehydrated through a series of ethanol gradients. The process was followed by antigen retrieval; the sections were heated to 110 °C in 1× antigen retrieval solution, made of diluted Citrate buffer (pH 6.0, 10×, Sigma-Aldrich, C9999-1000ML, Stockholm, Sweden), in a pressure cooker for 5 min. Non-specific epitopes were blocked using TBS-Tween 0.1%, supplemented with 5% goat serum (Sigma-Aldrich, G9023-5ML, Stockholm, Sweden) or horse serum (Sigma-Aldrich, H1270-5ML, Stockholm, Sweden) for 60 min. Afterwards, endogenous peroxidase activity was blocked with BOXALL^®^ peroxidase blocking solution for 10 min (Vector Laboratories, SP-6000, Newark, CA, USA), and the sections were incubated in primary antibody at 4 °C overnight. The following day, sections were incubated with the ImmPRESS^®^ HRP Goat Anti-Rabbit IgG Polymer Detection Kit, Peroxidase (Vector Laboratories, MP-7451, Newark, CA, USA), or the ImmPRESS^®^ HRP Horse Anti-Mouse IgG Polymer Detection Kit, Peroxidase (MP-7402), for 30 min, and stained with the ImmPACT^®^ DAB Substrate Kit, Peroxidase (HRP) (Vector Laboratories, SK-4105, Newark, CA, USA), for 1–2 min. Nuclear counterstaining was performed using Mayer’s hematoxylin, the sections were dehydrated with serial ethanol dilutions and xylene, and they were mounted with SignalStain^®^ Mounting Medium (Cell Signaling Technology, #14177, Leiden, The Netherlands). Imaging of the sections was performed using the ZEISS Axioscan 7 slide scanner at a magnification of 20×.

### 2.7. Cell Aggregation Quantification

D425 and D458 cell lines, treated with BRQ at IC50 or DMSO (0.01%), were imaged after 24, 48 and 72 h of treatment at 10× magnification using the Nikon Eclipse Ts2 inverted microscope. Cell aggregates (*n* = 3 biological replicates) consisting of >3 cells and large cell aggregates consisting of >15 cells were counted and quantified according to the percentage of large aggregates over the total number of aggregates for each time point across the 2 conditions. Statistical analysis was performed using two-way ANOVA with Bonferroni’s multiple comparisons test.

### 2.8. Statistical Analysis

GraphPad Prism version 10.1.1 for Mac (GraphPad Software, San Diego, CA, USA) was used for statistical analyses and graphs. IC50 (inhibitory concentration 50%) values were calculated using non-linear regression on concentration effect curves (model: Y = 100/(1 + (IC50/X)^HillSlope^)). Dunnett’s multiple comparison test was used to test for statistical significance by comparing two groups.

## 3. Results

### 3.1. DHODH Is Highly Expressed in MB and Correlates with MYC Expression

To explore the importance of DHODH in MB, mRNA expression levels of *DHODH* were assessed using publicly available MB datasets and compared across the four molecular subgroups. Within the Cavalli dataset, the shh subgroup exhibits the highest expression of *DHODH,* followed by the group 3 subgroup (Figure 1A). Using a further sub-division into 12 MB subtypes, *DHODH* was most highly expressed in group 3 gamma, shh alpha, and shh delta (Figure 1B). Furthermore, higher *DHODH* expression was significantly associated with worse overall survival in patients with group 3 MB in two datasets (Figure 1C,D). No correlation was found for the shh, wnt, and group 4 subgroup (Williamson, Cavalli) (Appendix A).

Patients with group 3 MB have the worst overall survival and remain a therapeutic challenge, in part due to the prevalence of metastatic disease [24]. Several studies have found *MYC* or *MYCN* amplifications to be common aberrations in this subgroup and have suggested that they might have overlapping functions [5,25,26]. Hence, we investigated whether there is a correlation between *DHODH* and *MYC* expression. There was a significant positive correlation between *DHODH* and *MYC* expression in the group 3 subgroup (Figure 1E,F) using the Cavalli dataset. This finding was validated in both non-*MYC*- and *MYC*-amplified group 3 MB using the Williamson dataset (Figure 1G and Appendix A). Based on these findings, we hypothesised that group 3 MB could be sensitive to DHODH inhibition, and that DHODH inhibition may affect *MYC* expression.

### 3.2. MB Cell Lines Are Sensitive to DHODH Inhibition

We sought to investigate whether MB cell lines are sensitive to DHODH inhibition. Within the publicly available CCLE dataset, *DHODH* was expressed at higher levels in MB, neuroblastoma, and rhabdoid tumour cell lines when compared to other paediatric cancer cell lines (Figure 2A). The MB cell lines also showed high expression of *MYC*, particularly the group 3 *MYC*-amplified D425 and D458 cell lines, as well as the group 4 D283 Med cell line (Figure 2B).

To investigate the effects of BRQ on MB, we assessed the viability of seven MB lines after a 72 h exposure to BRQ. The seven lines were derived from different subgroups, including two pairs of cell lines derived from primary tumours and metastasis at relapse from the same patients, as well as the patient-derived cell line MB-LU-181 [16]. BRQ demonstrated a concentration-dependent suppression of tumour cell viability, with IC50 values in the low nanomolar range (Figure 2C,D). The group 3 cell line Med8A and shh cell line DAOY exhibited increased sensitivity to BRQ, with IC50 values of 4.9 nM and 4.13 nM, respectively (Figure 2C,D). In the pair of group 4 MB cells, CHLA-01-MED and CHLA-01R-MED, BRQ was very effective in inhibiting cell viability in both the relapse/metastatic cells (CHLA-01R-MED, IC50 = 6.3 nM) and primary cells (CHLA-01-MED, IC50 = 2.97 nM) (Figure 2D). Similarly, in the group 3 MB cell line pair, D425 and D458, BRQ exhibited a lower IC50 value in the primary cell line, D425 (IC50 = 4.3 nM), compared to the relapsed/metastatic cell line, D458 (IC50 = 15 nM). (Figure 2C). Furthermore, an increase in cleaved PARP and cleaved Caspase 3, both indicators of cell death, was observed in D425 and D458 cells treated with BRQ (Appendix A). In addition, we observed that the two suspension cell lines exhibited a decreased capacity to form big aggregates in the presence of BRQ compared to the vehicle (DMSO) treatment (Figure 2E) and quantified in (S2D). MB-LU-181 was the most resistant cell line to BRQ, with an IC50 value of 87 nM.

### 3.3. BRQ Treatment Inhibits MYC Expression in Group 3 MB

Building on the observations that *DHODH* expression correlates with *MYC* expression in group 3 MB (Figure 1F) and the increased sensitivity of the *MYC*-amplified group 3 cell lines, D425 and D458, to BRQ treatment (Figure 2C), we evaluated the effects of BRQ treatment on the transcriptome of the matched cell lines. Gene expression analysis by RNA sequencing was performed on the D425 and D458 cell lines after 24 h and 72 h exposures to IC50 BRQ (D425 = 4.3 nM, D458 = 15 nM) or DMSO-treated control cells. Gene set enrichment analysis (GSEA) of hallmark gene sets, representing well-defined biological processes, demonstrated that eight and twenty-two gene sets were significantly downregulated (FDR qvalue < 0.25) in the D425 cell line after 24 h and 72 h BRQ treatment, respectively, and no gene sets were significantly upregulated (Appendix A). In the D458 cell line, nine and twenty-five gene sets were significantly downregulated after 24 h and 72 h BRQ treatment, respectively, while one upregulated gene set was identified after 24 h of treatment (Appendix A).

Among the top pathways downregulated by BRQ after 24 h in both cell lines were *MYC* targets V1 and V2, followed by the E2F and G2M checkpoints, and the MTORC1 pathways in the Hallmarks library (Figure 3A,B, Appendix A). GSEA charts and heatmaps for differential gene expression (DGE) in *MYC* targets V1 and V2 are illustrated as examples in (Figure 3C,D) for D425 and (Figure 3E,F) for D458. We observed significant downregulation of *MYC* expression and *ODC1* expression, which are known to be involved in nucleotide biosynthesis via the production of polyamines (Figure 3D) [27]. Moreover, in D458, the upstream *CAD* gene, known to perform the first three rate-limiting steps of pyrimidine biosynthesis, was also downregulated (Figure 3E). The inhibitory effect of BRQ on *MYC* target expression was maintained in both cell lines after 72 h of treatment, and the D425 cells also exhibited maintained inhibition of the *MYC* gene (Figure 3D,F). These results demonstrate the capacity of BRQ for sustained inhibition of *MYC* gene expression and its targets in *MYC*-amplified group 3 MB cell lines for at least 72 h. The number of shared and exclusive differentially expressed genes (DEGs) from our RNA-seq data in both cell lines, treated with BRQ for 24 h and 72 h, respectively, are represented in Venn diagrams (Figure 3G,H).

### 3.4. BRQ Inhibits Tumour Growth in a Zebrafish Xenograft MB Model

Following the promising in vitro results, we evaluated the in vivo potential of BRQ by studying tumour growth inhibition in the context of a zebrafish xenograft model of group 3 MB. Briefly, D425 and D458 cells tagged with green fluorescent protein (GFP) were transplanted into 1K-stage zebrafish embryos, which were monitored until 24 h post-fertilization (hpf), at which point successfully transplanted embryos were selected. The embryos were imaged at 24 hpf to track tumour cell localization, and then received a single-dose treatment of BRQ or vehicle control (DMSO) at non-toxic concentrations, following which they were imaged again after 72 h of treatment (96 hpf embryos). To automate the quantification process of tumour growth, we developed a custom image-based analysis script that can accurately distinguish and measure GFP-positive objects (Figure 4A).

The homing mechanisms of the transplanted GFP-labelled D425 and D458 cells were quantified to address whether there was preferential localization of the human cell lines in the brain area of the embryos. We employed a pipeline where transplanted embryos are imaged at 48 hpf and divided into five areas, including the hindbrain, mid/forebrain, pericardium, yolk sac, and the body/tail area, to track tumour cell localization according to the GFP signal [23]. Transplantation of D425 cells resulted in preferential localization in the brain area with 68% (hindbrain = 27%, mid/forebrain = 41%), 22% in the yolk sack, and 7% and 2% in the pericardium and body/tail, respectively (Figure 4B and Appendix A). Similarly, 56.9% of D458 cells localised in the brain area of embryos (hindbrain = 12%, mid/forebrain = 44.9%), with 31% homing in the yolk sac, 7.2% in the pericardium, and 4.9% in the body/tail area. The confirmation of preferential homing of the two transplanted cell lines to the brain area makes the zebrafish embryo xenograft a promising in vivo model for studying MB tumour progression.

We determined the tolerance to BRQ by exposing 24 hpf zebrafish embryos to increasing concentrations of BRQ for a total of 72 h, and simultaneously monitored toxicity. Out of the eight tested BRQ concentrations, seven showed no toxicity compared to the E3 medium (normal conditions), whilst the DMSO and BRQ 5 nM group showed slight toxicity signs, with one to two embryos per group presenting with cardiac oedema after 72 h of exposure, whereas the highest concentration of 10 nM showed toxicity starting from 24 h of exposure (Figure 4C). Therefore, we performed experiments with 2.5 nM BRQ, which was the highest concentration exhibiting no signs of toxicity.

Furthermore, 24 hpf zebrafish xenografts transplanted with either D425 or D458 cells were treated for 72 h with 2.5 nM BRQ or DMSO. Images were taken before the administration of the treatment at 24 hpf and after the 72 h of exposure treatment (at 96 hpf embryos), and each embryo was scored for an increase in the GFP signal using our custom image-based analysis script. BRQ significantly inhibited tumour growth of D458 xenografts compared to DMSO-treated control fish (*p* = 0.0028, Dunnett’s multiple comparison test) after 72 h of exposure, whereas the effect was not as pronounced for D425 xenografts (*p* = 0.2537, Dunnett’s multiple comparison test) (Figure 4D,E). Furthermore, a decrease in Ki67-positive cells and an increase in Caspase-3 positive cells were observed in the transplanted cells for both cell lines when treated with BRQ (Appendix A).

## 4. Discussion

The survival of patients with MB has significantly improved due to risk-adapting and multimodal therapies. However, metastatic spread of the disease occurs in 20–40% of newly diagnosed patients and is one of the leading causes of treatment failure [28,29,30,31]. In particular, group 3 MB tumours have a poor prognosis and demonstrate the highest prevalence of metastasis at initial diagnosis and recurrence [28]. In order to prevent or treat metastatic disease, patients often receive therapies with severe toxicities, which often results in long-term side-effects. Hence, a deeper understanding of MB biology and more therapies targeted at metastatic MB are urgently needed.

Dihydroorotate dehydrogenase (DHODH) is a flavoenzyme located in the inner mitochondrial membrane that is essential in de novo pyrimidine biosynthesis [32,33,34]. The pathway of de novo pyrimidine biosynthesis comprises six enzymatic reactions, conserved among species, the fourth of which is catalysed by DHODH [34]. In this reaction chain, after dihydroorotate enters the mitochondria, DHODH converts it to orotate by oxidation. Additionally, DHODH participates in the electron transport chain by mediating electron transfers between DHO and complex III [32]. Among DHODH inhibitors, the synthetic quinoline carboxylic acid BRQ exhibits selectivity and potency in a water-soluble molecule [35].

Amplification of the *MYC* gene is a distinguishable feature of group 3 MBs, occurring in 17% of patients, while it rarely occurs in patients of the other subgroups [36]. Moreover, group 3 MB patients presenting with *MYC* amplification have an increased risk of metastasis and recurrence [37,38]. In our study, we reveal the interplay of *MYC* and *DHODH* in group 3 MB, based on the group-specific expression of *DHODH* in MB tumours, and discover the sustained inhibition of *MYC* expression upon BRQ single-dose treatment in two human MB cell lines. Within public datasets, increased *DHODH* expression was identified in group 3 and shh MB tumours and was associated with poor survival in group 3 (Figure 1A–D). This group-specific *DHODH* expression, coupled with the correlation between *MYC* and *DHODH* expression in the same groups (Figure 1E,F and Appendix A), could indicate that group 3 and shh MB patients could potentially benefit from DHODH inhibition. We, therefore, treated the D425 primary and D458 metastatic group 3 MB cell line pair with a single-dose treatment of the DHODH inhibitor BRQ, and compared the bulk RNA expression profiles of the cells 24 and 72 h after treatment to detect whether the effect of BRQ would be sustained over time. Our results demonstrated maintained downregulation of *MYC* targets V1 and V2 in the Hallmarks library for both cell lines, except for V2 targets after 72 h of D458 treatment, which was not statistically significant (Figure 3A,B, Appendix A). Additionally, E2F, the G2M checkpoint, and MTORC1 pathways were downregulated upon BRQ treatment, with all exhibiting a tendency for recovery in the D458 cell line after 72 h of treatment, with the exception of the G2M checkpoint pathway. We suspect that the difference between the two cell lines can be attributed to the metastatic nature of D458 [39]. Finally, the upregulated pathways, based on the Hallmarks library, in both cell lines and timepoints included pathways related to metabolism, as well as the interferon–gamma response and apoptosis, in the first 24 h upon treatment, which could be an indication of cell death caused by the treatment of BRQ with the IC50 concentration.

Inhibition of DHODH has been identified as a promising therapeutic strategy for various cancer types, given the significance of cellular metabolism deregulation as a hallmark of cancer [40]. DHODH inhibition is believed to have a double negative effect on cancer cells, by impairing cell growth, due to insufficient de novo pyrimidine biosynthesis, as well as by aiding cell differentiation, potentially through elongation of transcription [41]. To date, the use of BRQ has not shown positive results in advanced stages of solid tumours, including breast, colon, lung, and prostate cancers in the context and doses applied in phase 1 and 2 clinical trials [42,43,44,45,46,47,48]. However, BRQ has exhibited promising anti-cancer activity targeting DHODH, as recently elucidated in preclinical anti-cancer studies in AML [41] and neuroblastoma [49], leading to the establishment of an ongoing clinical trial (ClinicalTrials.gov identifier: NCT03760666) studying the effect and tolerability of BRQ against AML in adults. Additionally, DHODH was also newly recognised as a potential targetable candidate in *MYC*-amplified group 3 MB [12]. In contrast to our study, Gwynne et al. selected the DHODH inhibitor BAY2402234 for their in vivo orthotopic PDX experiments, due to the lesser capacity of the latter to penetrate the blood–brain barrier (BBB) [12].

Our study highlights the zebrafish xenograft assay as a valuable orthotopic model system for rapid in vivo studies of MB. Transplantation of group 3 MB cell lines D425 and D458 in 1K-stage zebrafish embryos demonstrated preferential localization to the brain area, and, in particular, the mid/forebrain (Figure 4B and Appendix A). Importantly, we demonstrate that patient-derived cells can successfully engraft and grow in the zebrafish model, thus allowing for fast testing and expansion of primary patient samples. In this study, we used the zebrafish xenograft assay to investigate the drug responses of the transplanted group 3 cell lines to BRQ, and found significantly inhibited tumour growth and an increase in Caspase-3 positive cells in D458 xenografts compared to vehicle (DMSO)-treated controls (Figure 4D,E and Appendix A) after 72 h of exposure. The potential and robustness of the zebrafish xenograft model have also recently been demonstrated using shh and group 4 MB cell lines by van Bree et al. [23]. Hence, this model can create opportunities for personalised drug sensitivity screens, allowing for rapid testing that can provide personalised treatment strategies for individual patients.

## 5. Conclusions

We demonstrate that DHODH is a druggable target in group 3 MB, and that DHODH inhibition may represent a therapeutic opportunity for patients with group 3 MB. Our established zebrafish xenograft model is a valuable orthotopic model that can be used for fast in vivo drug testing of MB.

## Figures and Tables

**Figure 1 cancers-16-04162-f001:**
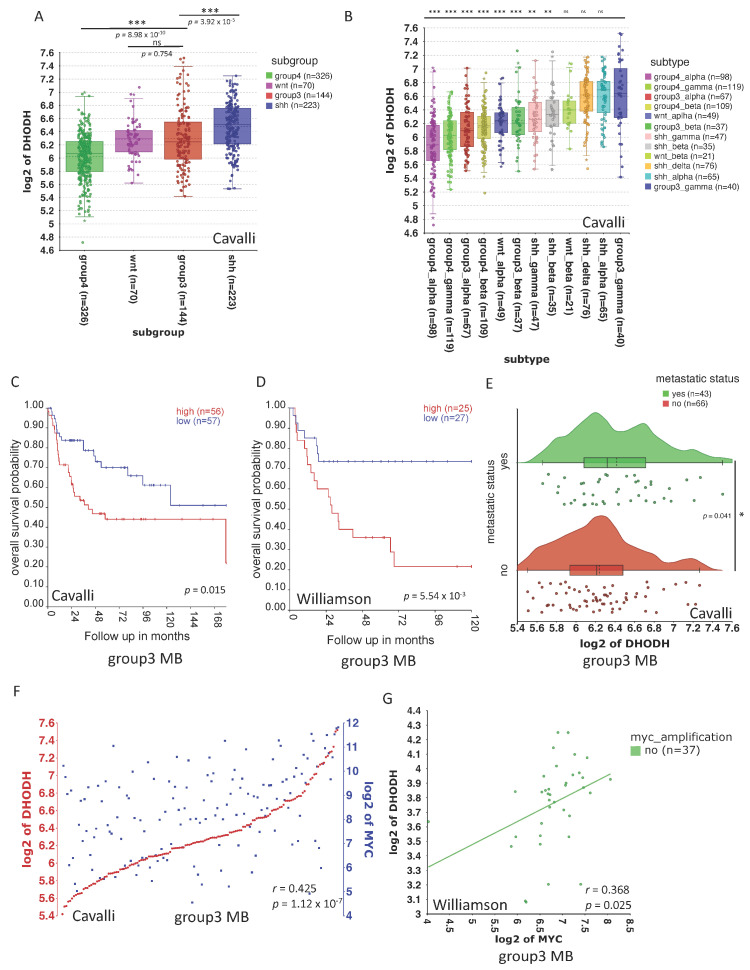
*DHODH* is expressed in MB. (**A**,**B**) mRNA expression of *DHODH* in 763 patients with MB (Cavalli dataset) stratified into molecular subgroups (wnt, shh, group 3, and group 4) and molecular subtypes (shh alpha, shh beta, shh gamma, shh delta, wnt alpha, wnt beta, group 3 alpha, group 3 beta, group 3 gamma, group 4 alpha, group 4 beta, group 4 gamma). For the cohort shown in A, n = 763 primary MBs, including group 3, n = 144; group 4, n = 326; shh, n = 223; and wnt, n = 70. For the cohort shown in B, n = 763, primary MBs including group 4 alpha, n = 98 (*p* = 9.68 × 10^−11^); group 4 gamma, n = 119 (*p* = 2.49 × 10^−8^); group 3 alpha, n = 67 (*p* = 4.11 × 10^−6^); group 4 beta, n = 109 (*p* = 2.56 × 10^−6^); wnt alpha, n = 49 (*p* = 9.99 × 10^−5^); group 3 beta, n = 37 (*p* = 7.99 × 10^−4^); shh gamma, n = 47 (*p* = 1.09 × 10^−3^); shh beta, n = 35 (*p* = 8.99 × 10^−3^); wnt beta, n = 21 (*p* = 0.051); shh delta, n = 76 (*p* = 0.778); shh alpha, n = 65 (*p* = 0.892); and group 3 gamma, n = 40. *p* values from one-way ANOVA across the four MB subgroups or subtypes, respectively, are shown. * *p* < 0.05, ** *p* < 0.01, *** *p* < 0.001 (**C**,**D**) The overall survival of group 3 MB patients separated by median *DHODH* expression in two publicly available datasets. For the cohort shown in C, the Williamson dataset was used with n = 331 primary MBs, including group 3, n = 52. For the cohort shown in D, the Cavalli dataset was used with n = 763 primary MBs, including group 3, n = 113. Red, *DHODH* above median (high); blue, *DHODH* below median (low). Groups are compared using the log-rank test. (**E**) A raincloud plot showing the average mRNA expression of *DHODH* in group 3 medulloblastoma patients stratified according to metastatic disease status using the Cavalli dataset. (**F**) The correlation of *DHODH* with *MYC* in group 3 medulloblastoma patients using the Cavalli dataset, n = 144, group 3 primary medulloblastomas. (**G**) The correlation of *DHODH* with *MYC* in group 3 medulloblastoma patients showing no *MYC* amplification, using the Williamson dataset, n = 37, group 3 primary medulloblastomas with no *MYC* amplification.

**Figure 2 cancers-16-04162-f002:**
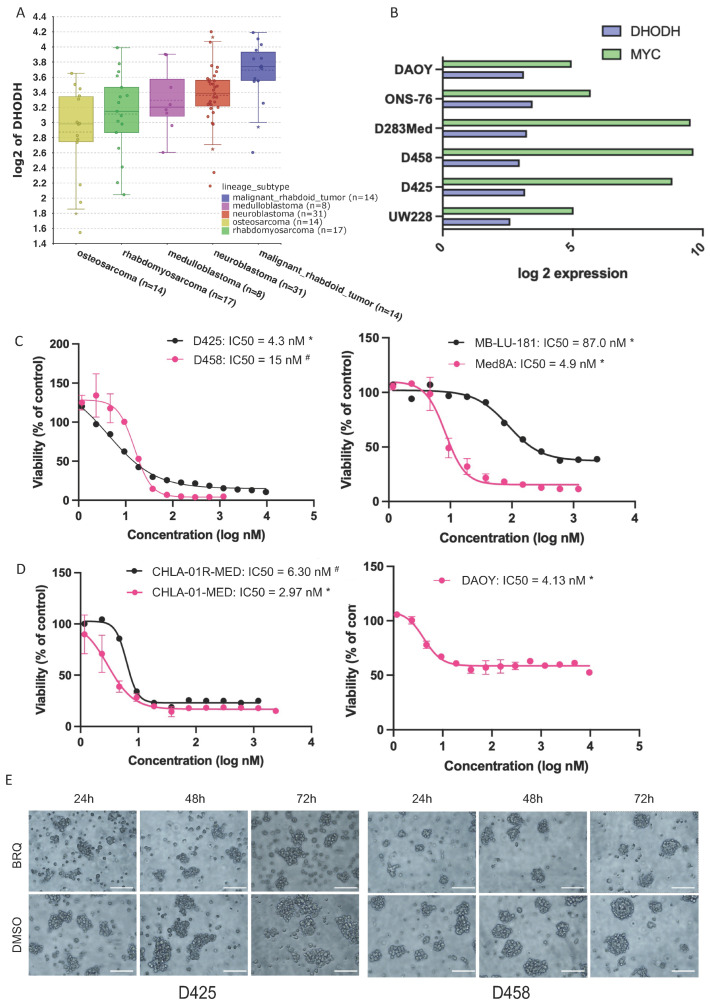
MB cell lines express *DHODH* and are sensitive to pharmacological treatment with BRQ. (**A**) *DHODH* expression in paediatric tumour cell lines of the CCLE dataset, grouped by primary disease. (**B**) *DHODH* and *MYC* expression in medulloblastoma cell lines of the CCLE dataset. (**C**,**D**) Dose-response curves for cell viability (assessed by WST1 assay) after 72 h of BRQ treatment in a panel of seven medulloblastoma cell lines. shh: DAOY, Group 3: D425, D458, MB-LU-181, Med8A; and Group 4: CHLA-01-MED, CHLA-01R-MED. Cell lines marked as primary * or metastatic #. (**E**) Examples of compromised tumour aggregate formation of the suspension cell lines (**E**) D425 and D458 after pharmacological treatment with BRQ IC50 for 72 h. Cells were images at 10× magnification. Scale bars, 100 μm.

**Figure 3 cancers-16-04162-f003:**
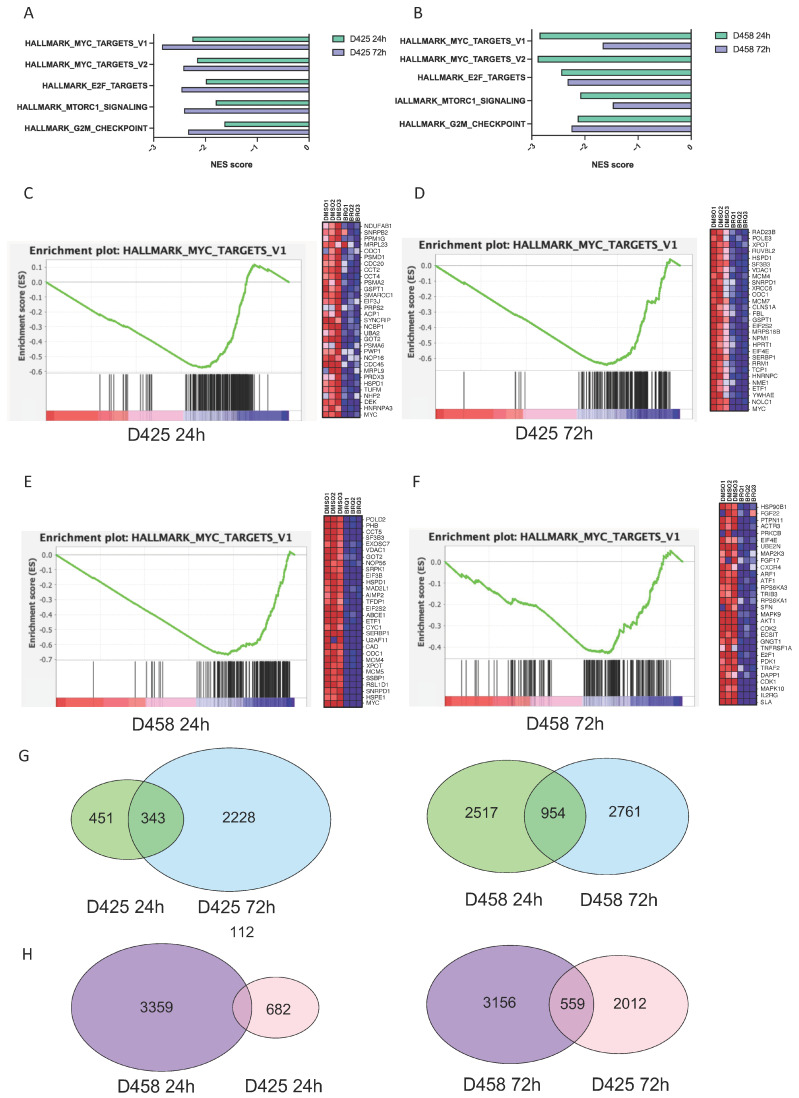
BRQ treatment has a sustained inhibition of MYC target expression. (**A**,**B**) Normalised enrichment scores (NESs) of significantly downregulated gene sets using gene set enrichment analysis (GSEA) of RNA-Seq data in BRQ-treated (single-treatment with IC50) cell lines D425 and D458 after 24 h and 72 h treatment with single-treatment of BRQ IC50. Gene set names from MSigDB (Hallmarks and CGP) are provided on the y axis. Negative NES indicates significant downregulation. (**C**–**F**) NESs of significantly deregulated MYC target genes from the GSEA of RNA-Seq data in BRQ-treated medulloblastoma cell lines D425 and D458. Left panels: GSEA enrichment plot demonstrating MYC target gene downregulation (MSigDB Hallmarks MYC target v1 gene set) in BRQ-treated cell lines, which is maintained at 24 h after 1 dose of BRQ (for C: NES = −2.25, NOM P = 0.000, FDR q = 0.000; for D: NES = −2.84, NOM P = 0.000, FDR q = 0.000; for E: NES = −2.86, NOM P = 0.000, FDR q = 0.000; for F: NES = −1.67, NOM P = 0.000, FDR q = 0.004). Right panels: heatmap enrichment scores of top deregulated MYC target genes in BRQ-treated cell lines at 72 h after 1 dose of BRQ. (**G**,**H**) Venn diagrams of differentially expressed genes (DEGs) in BRQ-treated MB cell lines D425 and D458. (**G**) After 24 h and 72 h treatment with single-treatment of BRQ IC50. (**H**) DEGs according to treatment timepoint (24 h vs. 72 h) and according to cell line (D425 vs. D458) (for (**G**,**H**): Adj *p* < 0.05).

**Figure 4 cancers-16-04162-f004:**
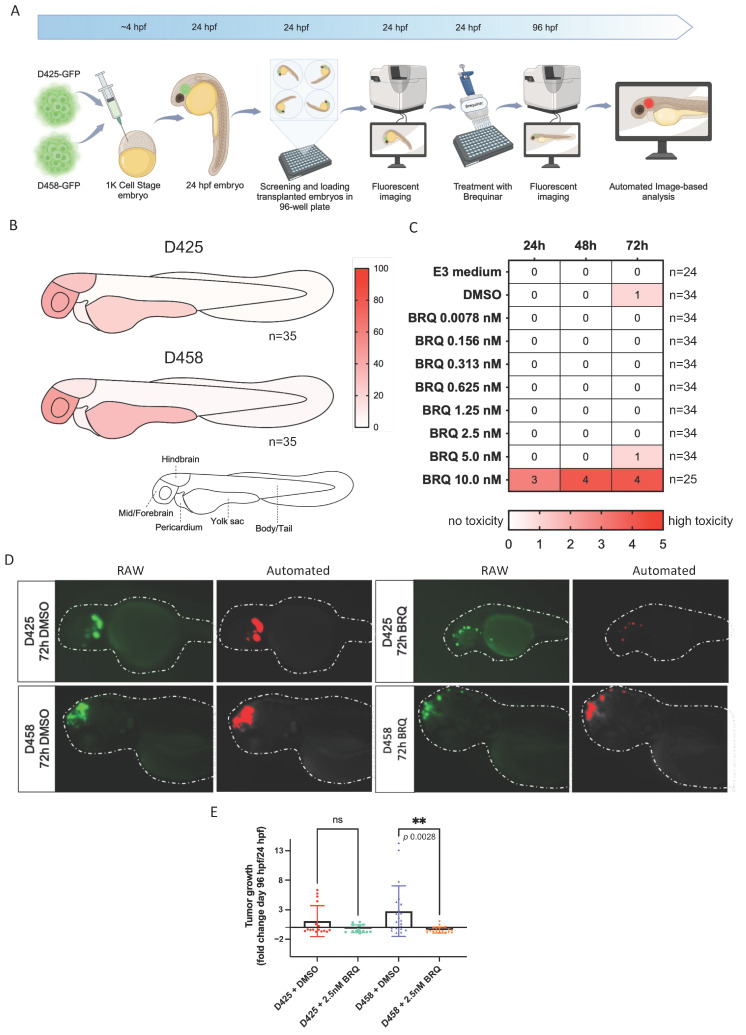
BRQ treatment inhibits tumour growth in a group 3 MB zebrafish xenograft model. (**A**) Schematic overview of group 3 MB xenografts in zebrafish. D425 and D458 cell lines were transduced with green fluorescent protein (GFP), FACS-sorted, and injected into the 1K cell stage of zebrafish embryos, monitored until 24 h post-fertilization (hpf) for successful transplantation, and selected for exposure to treatments. Zebrafish embryos transplanted with either D425 or D458 were treated with vehicle or BRQ (2.5 nM) for 72 h, and tumour areas were imaged before treatment (24 hpf) and after 72 h treatment (96 hpf). (**B**) Schematic heatmaps of the tumour cell location of transplanted group 3 cell lines in 48 hpf old embryos. 1 = hindbrain area; 2 = mid/forebrain area; 3 = pericardium area; 4 = yolk sac area; 5 = body and tail area. (**C**) 24 hpf zebrafish embryos were exposed to sequentially increasing BRQ concentrations to deduce maximum tolerance. Toxicity was noted at 24 h, 48 h, and 72 h after single-treatment with BRQ, DMSO, or E3 medium, and scored between 0 (no toxicity, white) and 5 (instant, lethal toxicity, red). (**D**) Representative images of included transplanted zebrafish embryos after 72 h treatment with BRQ or vehicle (DMSO). Left panel: raw images showing homing of transplanted cell lines in green. Right panel: accurate identification of transplanted cells by our automated image-based pipeline in red. (**E**) Tumour growth from automated image-based analyses in transplanted zebrafish embryos, expressed as fold-change 96 hpf/24 hpf. The mean fold-change ± SD is presented (D425 + BRQ 2.5 nM n = 16, D425 + DMSO n = 15, D458 + BRQ 2.5 nM n = 21, D458 + DMSO n = 18). One-way ANOVA showed significant growth inhibition, only for D458, after treatment with 2.5 nM BRQ, *p* = 0.0028 (comparisons with vehicle for D425, *p* = 0.2537). ** *p* < 0.01.

## Data Availability

Data available upon request.

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
