# Peer review of "DHODH Inhibition Suppresses *MYC* and Inhibits the Growth of Medulloblastoma in a Novel In Vivo Zebrafish Model"

_cancers, 2024, doi:10.3390/cancers16244162_

Round 1

Reviewer 1 Report

Comments and Suggestions for Authors

In this study, Tsea and colleagues examine the effects of brequinar (BRQ), an inhibitor of DHODH, on aggressive group 3 medulloblastoma (MB). The authors first explored the public dataset and demonstrated that DHODH is highly expressed in group 3 MB which also correlated with high MYC expression. Using different MB cell lines, the authors showed BRQ can inhibit cell growth in vitro. In addition, RNA-seq results indicated that BRQ treatment inhibit the expression of MYC and its targets. Lastly, the authors developed a zebrafish xenograft MB model and demonstrated that BRQ can inhibit tumor growth in vivo. Overall, the manuscript is well-written and results largely support the authors’ conclusions. However, there are a few issues needed to be addressed.

1.     In Fig.2B, there does not seem to be a correlation between DHODH and MYC expression in those cell lines.

2.     The authors need to distinguish the effects of BRQ on MB cells. Does it inhibit cell proliferation , induce cell death or both?

3.     Fig.2E and 2F needs quantification.

4.     For the in vivo experiment, it would be great to include a cell line from a different subtype to demonstrate the specificity to group 3 MB.

Author Response

Reviewer Comments:

Reviewer 1:

Please see the attached word for references to relevant figures now added to the manuscript.

In this study, Tsea and colleagues examine the effects of brequinar (BRQ), an inhibitor of DHODH, on aggressive group 3 medulloblastoma (MB). The authors first explored the public dataset and demonstrated that DHODH is highly expressed in group 3 MB which also correlated with high MYC expression. Using different MB cell lines, the authors showed BRQ can inhibit cell growth in vitro. In addition, RNA-seq results indicated that BRQ treatment inhibit the expression of MYC and its targets. Lastly, the authors developed a zebrafish xenograft MB model and demonstrated that BRQ can inhibit tumor growth in vivo. Overall, the manuscript is well-written and results largely support the authors’ conclusions. However, there are a few issues needed to be addressed.

  1. In Fig.2B, there does not seem to be a correlation between DHODH and MYC expression in those cell lines.

We appreciate and agree with the reviewer’s comment. In figure 2B there is no correlation between the cell lines. However, we think the limitation of cell lines (n=6) might be an explanation.

  1. The authors need to distinguish the effects of BRQ on MB cells. Does it inhibit cell proliferation, induce cell death or both?

We thank the reviewer for this comment. To address this concern, we have performed additional experiments using Western blots assessing cleaved PARP (as a marker of cells undergoing apoptosis) in D425 and D458 cells after single treatment of BREQ (24 h, 72 h). The data is added to figure S. 2A (see also figure below). In the revised manuscript we have also included densitometric quantifications of Western blot bands as figure S.2B (see also figure below). We have also included immunohistochemistry staining for proliferating cells and apoptosis/cell death in D425 and D458 cell lines after 24 h or 72 h treatment with BRQ IC50 treatment or DMSO. This data is now added as figure S. 2E (see also figure below).

See: Figure S.2A. Cleaved PARP expression in D425 and D458 cells after 24h and 72h treatment with single treatment with BRQ.

See: Figure S.2B. Densitometric analyses of protein bands from Western blots, from at least three biological replicates

See: Figure S.2E. Immunohistochemistry staining for proliferating cells (Ki-67) and apoptosis/cell death. (see below)

  1. 2E and 2F needs quantification.

We quantified cells grown and treated as figure 2 E-F by counting cell aggregates with more than 15 cells (24 h, 48 h, 72 h). This data is now added as figure S.2D (see also figure below). Medulloblastoma growth, assessed as cell aggregation confluence, was more efficiently repressed in both cell lines treated with BREQ, suggesting an involvement of proliferation impairment or other types of cell death in addition to apoptosis.

See: Figure S.2D. Quantification of D425 and D458 cellular aggregation at 24 h, 48 h and 72h after single-treatment with BRQ IC50.

  1. For the in vivo experiment, it would be great to include a cell line from a different subtype to demonstrate the specificity to group 3 MB.

We agree with the reviewer that it is highly interesting to demonstrate the specificity to group 3 MB. That in vivo experiment including cell lines from different subtypes has been evaluated before by van Bree et al., In this study, the authors show that medulloblastoma cell lines from different subtypes all home toward the hindbrain region when neural stem cell conditions are present. They were all transplanted into the blastula stage of the embryo. It appears that this can be a global MB mechanism and not specific to group 3 MB. (van Bree et al., 2024)

van Bree, N., Oppelt, A. S., Lindstrom, S., Zhou, L., Boutin, L., Coyle, B., . . . Wilhelm, M. (2024). Development of an orthotopic medulloblastoma zebrafish model for rapid drug testing. Neuro Oncol. doi:10.1093/neuonc/noae210

Reviewer 2 Report

Comments and Suggestions for Authors

The authors have tested effects of BRQ (Brequinar), a DHODH inhibitor on MYC levels and  hence growth of pediatric medulloblastoma tumor. The study is well thought of, conducted and presented. I have few minor comments listed below:

1. The introduction can include more details on group 3 medulloblastoma patients. What are the signaling pathways dysregulated? And how often is metastasis observed? And at what point does metastasis commonly occus? Dysregulated pathways in primary vs secondary tumors etc.

2. The authors should explain origin of 7 MB cell lines in more detail. This will be useful to interpret the results. It will also be useful to mark in the graph which cell lines are from primary tumors and which are from metastatic ones. 

3. Venn diagrams showing numbers of differentially regulated genes in both cell lines (overlap and distinct) from their RNA-seq data will be good to have. 

4. The authors should include a graphical model in the end that summarizes their findings.  

Author Response

Reviewer Comments:

Reviewer 2:

Please see the attached word for references in figures newly added to the manuscript

The authors have tested effects of BRQ (Brequinar), a DHODH inhibitor on MYC levels and hence growth of pediatric medulloblastoma tumor. The study is well thought of, conducted and presented. I have few minor comments listed below:

  1. The introduction can include more details on group 3 medulloblastoma patients. What are the signaling pathways dysregulated? And how often is metastasis observed? And at what point does metastasis commonly occurs? Dysregulated pathways in primary vs secondary tumors etc.

We thank the reviewer for this comment. In the revised manuscript we have in the introduction included more details on group 3 MB, as suggested (see line 46-55 in the revised manuscript).

  1. The authors should explain origin of 7 MB cell lines in more detail. This will be useful to interpret the results. It will also be useful to mark in the graph which cell lines are from primary tumors and which are from metastatic ones. 

We agree with the reviewer that information regarding the cell lines is of importance to interpret results. All cell lines, along with growth patterns, group classification and other characteristics, are listed in S. table 1. Which cell lines that are from primary tumors and which are from metastatic tumors are marked with * and # in an updated version of supplementary table 1. In figure 2 C-D primary cell lines are marked with * and metastatic cell lines are marked with #.

  1. Venn diagrams showing numbers of differentially regulated genes in both cell lines (overlap and distinct) from their RNA-seq data will be good to have. 

We appreciate the reviewer’s comment and have in the revised manuscript added Venn diagrams presenting differentially expressed genes (DEGs) from our RNA-seq data in both cell lines, treated with BREQ for 24 h and 72 h, as figure 2 G-H (see also figure below).

See: Figure 2 G-H. Venn diagrams presenting differentially expressed genes.

We have also created Venn diagrams based on figure 2 C-F presenting genes overlapping in both cell lines, 24 h and 72 h (not added in manuscript, see figure below). We see an overlap of ODC1, HSPD1, MYC after 24 h of BREQ treatment and an overlap of EIF4E after 72 h of BREQ treatment.

  1. The authors should include a graphical model in the end that summarizes their findings. 

We appreciate this reviewer's comment and have attached a graphical model in the revised manuscript which summarizes our main findings.

Reviewer 3 Report

Comments and Suggestions for Authors

DHODH Inhibition Suppresses MYC and Inhibits the Growth of Medulloblastoma in a Novel in Vivo Zebrafish Model

Authors: Ioanna Tsea, Thale Kristin Olsen, Panagiotis Alkinoos Polychronopoulos, David B. Sykes, Ninib Baryawno and Cecilia Dyberg

General comment:

In this study the authors use the ZF model and in vitro cultures with human cell lines, to study the effect of Dihydroorotate dehydrogenase (DHODH) inhibition on Medulloblastoma treatment. The authors show that, by inhibiting DHODH, there is increased cytotoxicity in vitro, leading to reduced tumor cell growth and also reduction in MYC levels. Overall, the paper is well written and the results clearly presented, although some additional experiments should be done to confirm the conclusions presented by the authors.

Major comments

1.     Please clarify why, in Fig 2C, the spheroid-forming capacity was tested with and without the drug treatments? Also, couldn´t the authors instead test the drug sensitivity in already formed spheroids to see if the effects were similar?

2.     Why did the authors decide to inject tumor cells into 1k stage embryos instead of waiting until 2dpf and inject directly in the brain? Can’t the migration be affected by the treatments and that lead do confusing results?

3.     It is not clear by these results if treatment is inducing apoptosis or inhibiting proliferation? In addition, the authors state that “DHODH inhibition is believed to have a double negative effect on cancer cells, by impairing cell growth, due to insufficient de novo pyrimidine biosynthesis, as well as by aiding cell differentiation potentially through elongation of transcription.” However, in the results section, for the in vitro experiments – figure 2 – the authors show results concerning cell viability (cytotoxicity). I would like to know if the authors had the opportunity to analyze cell proliferation, instead, to show the specific effect of DHODH inhibition through impairment of cell growth in vitro?

4.     The same question should be answered for in vivo experiments. Could the authors please provide additional immunofluorescence staining’s to show in the tumor if there is reduction of proliferation and/or increased apoptosis?

5.     The authors start the discussion by saying that the highest cause of MB mortality is the occurrence of metastases. However, there were no results presented concerning the metastases in the zebrafish model. Did the authors evaluate the metastatic spread in the model?

6.     The term “inhibited tumor size” should be replaced by something more concrete. As it is, is not clear what the authors mean.

Minor comments:

1.     In the Introduction, Lines 44-45: the sentence is not clear – patients die from the primary tumor and not from the metastatic spread? Can the authors please clarify?

2.     In the methods, Lines 154-155 – “(…) 4 hpf embryo was injected with approximately 100-300 cells in the centre 154 of the cell mass according to van Bree et al 2024 (preprint)”. The injection proceedure should be better explained. It is now clear to which pre-printed paper is the author refering to.

3.     In the Results section, “Within the publicly available CCLE dataset we, DHODH was expressed at higher levels in MB, neuroblastoma and rhabdoid tumour cell lines when compared to other pediatric cancer cell lines (Fig 2A). “ This sentence is not clear. Could the authors please clarify?

4.     Line 335 – inhibited tumor size – shouln’t be inhibited tumor growth or reduced tumor size? It is not clear what the authors mean.

Author Response

Reviewer 3:

Major comments

Please see the attached word for references to relevant new figures added.

  1. Please clarify why, in Fig 2C, the spheroid-forming capacity was tested with and without the drug treatments? Also, couldn´t the authors instead test the drug sensitivity in already formed spheroids to see if the effects were similar?

We thank the reviewer for this comment. In the revised manuscript we have clarified that the cells grow in suspension as aggregates and not as spheroids.

  1. Why did the authors decide to inject tumor cells into 1k stage embryos instead of waiting until 2dpf and inject directly in the brain? Can’t the migration be affected by the treatments and that do confusing results?

We thank the reviewer for this highly relevant comment. We used the same           set up/protocol as van Bree et al., (van Bree et al., 2024) with injection of tumor cells into 1k stage zebrafish embryos. The migration seems to be to the same position in the hindbrain in both BREQ treated zebrafishes and DMSO treated zebrafishes.

We have used a standard setup for the zebrafish experiment with treatment for 72 h where no ethical permit is needed; according to the EU directive 2010/63/EU and the Swedish legislation on animal experimentation, a larva/zebrafish embryo needs to be freely feeding in order to be considered as model organism. This state corresponds to about 5 days (120 hours) post fertilization.

  1. It is not clear by these results if treatment is inducing apoptosis or inhibiting proliferation? In addition, the authors state that “DHODH inhibition is believed to have a double negative effect on cancer cells, by impairing cell growth, due to insufficient de novo pyrimidine biosynthesis, as well as by aiding cell differentiation potentially through elongation of transcription.” However, in the results section, for the in vitro experiments – figure 2 – the authors show results concerning cell viability (cytotoxicity). I would like to know if the authors had the opportunity to analyze cell proliferation, instead, to show the specific effect of DHODH inhibition through impairment of cell growth in vitro?

We agree with the reviewer that it’s important to dissect the mechanism of DHODH inhibition not only using cell viability assay. In the revised manuscript we have performed additional experiments using Western blots assessing cleaved PARP in D425 and D458 cells after treatments with BREQ (24 h, 72 h). The results are added to figure S. 2A (see also figure below). In the revised manuscript we have also included densitometric quantifications of Western blot bands as figure S. 2B (see also figure below). We also quantified figure 2 E-F by counting cell aggregates with more than 15 cells (24 h, 48 h, 72 h). This data is now added as figure S.2D (see also figure below). We have also included immunohistochemistry staining for proliferating cells and apoptosis/cell death in D425 and D458 cell lines after 24 h or 72 h treatment with BRQ IC50 treatment or DMSO. This data is now added as figure S. 2E (see also figure below).

See: Figure S.2A. Cleaved PARP expression in D425 and D458 cells after 24h and 72h treatment with single treatment with BRQ.

See: Figure S.2B. Densitometric analyses of protein bands from Western blots, from at least three biological replicates.

See: Figure S.2D. Quantification of D425 and D458 cellular aggregation at 24 h, 48 h and 72h after single-treatment with BRQ IC50.

See: Figure S.2E. Immunohistochemistry staining for proliferating cells (Ki-67) and apoptosis/cell death

  1. The same question should be answered for in vivo experiments. Could the authors please provide additional immunofluorescence staining’s to show in the tumor if there is reduction of proliferation and/or increased apoptosis?

As the reviewer suggest, we have performed immunofluorescence stainings on zebrafish sections from the in vivo experiment using the antibodies Ki67 (proliferation) and cleaved Caspase-3 (apoptosis). The stainings are now included in figure S.3 B-C (see also figure below) and figure S.3D. We observed an increase of cleaved Caspase-3 positive cells and a decrease of Ki67 positive cells in D458 zebrafishes treated with BREQ for 72 h hours compared to DMSO treated zebrafishes.

See: Figure S.3 B-C. Immunohistochemical analyses showing % of cells positive for proliferation marker KI-67 and Caspase-3 and in transplanted zebrafish embryos after 72h treatment with 2.5 nM BRQ treatment or DMSO.

  1. The authors start the discussion by saying that the highest cause of MB mortality is the occurrence of metastases. However, there were no results presented concerning the metastases in the zebrafish model. Did the authors evaluate the metastatic spread in the model?

We appreciate the reviewer’s interest in our zebrafish model. Unfortunately, we didn’t have the chance to evaluate the metastatic spread in this manuscript, due to ethical permit needed, but it would be very interesting to continue with a metastatic zebrafish model in future studies.

  1. The term “inhibited tumor size” should be replaced by something more concrete. As it is, is not clear what the authors mean.

We thank the reviewer for carefully reading the manuscript and have changed inhibited tumor size to inhibited tumor growth in the revised manuscript.

Minor comments:

  1. In the Introduction, Lines 44-45: the sentence is not clear – patients die from the primary tumor and not from the metastatic spread? Can the authors please clarify?

We thank the reviewer for this comment. The sentence- Patients with MB seldom die from the primary tumour, but rather because of metastatic disease is clarified with more background information regarding group 3 MB.

  1. In the methods, Lines 154-155 – “(…) 4 hpf embryo was injected with approximately 100-300 cells in the centre 154 of the cell mass according to van Bree et al 2024 (preprint)”. The injection proceedure should be better explained. It is now clear to which pre-printed paper is the author refering to.

We agree with the reviewer that the information should be better explained. The paper we are referring to is now published. (van Bree et al., 2024) . van Bree, N., Oppelt, A. S., Lindstrom, S., Zhou, L., Boutin, L., Coyle, B., . . . Wilhelm, M. (2024). Development of an orthotopic medulloblastoma zebrafish model for rapid drug testing. Neuro Oncol. doi:10.1093/neuonc/noae210

  1. In the Results section, “Within the publicly available CCLE dataset we, DHODH was expressed at higher levels in MB, neuroblastoma and rhabdoid tumour cell lines when compared to other pediatric cancer cell lines (Fig 2A). “ This sentence is not clear. Could the authors please clarify?

We thank the reviewer for this comment. We have taken away “we”. Hopefully the sentence is a bit clearer.

  1. Line 335 – inhibited tumor size – shouln’t be inhibited tumor growth or reduced tumor size? It is not clear what the authors mean.

We agree and have changed inhibited tumor size to inhibited tumor growth in the revised manuscript.

Round 2

Reviewer 1 Report

Comments and Suggestions for Authors

The manuscript has been improved with the revision.  I have no more comments regarding this study. 

Author Response

Reviewer comments: The manuscript has been improved with the revision.  I have no more comments regarding this study. 

Author response: We thank the reviewer for their time in reviewing this manuscript. 

Reviewer 3 Report

Comments and Suggestions for Authors

I would like to thank the authors for improving the manuscript and replying to the comments done.

There are only to points that need to be further addressed:

  1. Reviewer question: Why did the authors decide to inject tumor cells into 1k stage embryos instead of waiting until 2dpf and inject directly in the brain? Can’t the migration be affected by the treatments and that do confusing results?

Authors reply:

We thank the reviewer for this highly relevant comment. We used the same set up/protocol as van Bree et al., (van Bree et al., 2024) with injection of tumor cells into 1k stage zebrafish embryos. The migration seems to be to the same position in the hindbrain in both BREQ treated zebrafishes and DMSO treated zebrafishes.

We have used a standard setup for the zebrafish experiment with treatment for 72 h where no ethical permit is needed; according to the EU directive 2010/63/EU and the Swedish legislation on animal experimentation, a larva/zebrafish embryo needs to be freely feeding in order to be considered as model organism. This state corresponds to about 5 days (120 hours) post fertilization.

Reviewer question:

I thank the authors for the answer provided which, unfortunately does not completely answer my question. The treatment could be done for 72h still if the experiment was performed at 2dpf. At this time point, the cells could be directly injected in the ZF brain. Considering this, I would like to question again why the authors chose the van Bree et al., 2024 protocol instead of injecting directly in the brain of 2dpf embryos.

3.        Reviewer question: It is not clear by these results if treatment is inducing apoptosis or inhibiting proliferation? In addition, the authors state that “DHODH inhibition is believed to have a double negative effect on cancer cells, by impairing cell growth, due to insufficient de novo pyrimidine biosynthesis, as well as by aiding cell differentiation potentially through elongation of transcription.” However, in the results section, for the in vitro experiments – figure 2 – the authors show results concerning cell viability (cytotoxicity). I would like to know if the authors had the opportunity to analyze cell proliferation, instead, to show the specific effect of DHODH inhibition through impairment of cell growth in vitro?

Authors reply: We agree with the reviewer that it’s important to dissect the mechanism of DHODH inhibition not only using cell viability assay. In the revised manuscript we have performed additional experiments using Western blots assessing cleaved PARP in D425 and D458 cells after treatments with BREQ (24 h, 72 h). The results are added to figure S. 2A (see also figure below). In the revised manuscript we have also included densitometric quantifications of Western blot bands as figure S. 2B (see also figure below). We also quantified figure 2 E-F by counting cell aggregates with more than 15 cells (24 h, 48 h, 72 h). This data is now added as figure S.2D (see also figure below). We have also included immunohistochemistry staining for proliferating cells and apoptosis/cell death in D425 and D458 cell lines after 24 h or 72 h treatment with BRQ IC50 treatment or DMSO. This data is now added as figure S. 2E (see also figure below).

Reviewer question:

Thank you for providing the additional data. Unfortunately it is still not clear if the treatment is reducing cell proliferation since cleaved PARP allows the assessment of apoptosis and there is no quantification of proliferation in the images provided in figure S. 2E. Could the authors quantify the proliferation in controls vs treatment conditions and show if there is a difference or not?

Author Response

2. Reviewer question:

I thank the authors for the answer provided which, unfortunately does not completely answer my question. The treatment could be done for 72h still if the experiment was performed at 2dpf. At this time point, the cells could be directly injected in the ZF brain. Considering this, I would like to question again why the authors chose the van Bree et al., 2024 protocol instead of injecting directly in the brain of 2dpf embryos.

Authors reply: Thank you for allowing us to clarify the rationale behind the particular design of the zebrafish experiments. As the reviewer suggests, an alternative design of the animal experiments could be to directly inject the medulloblastoma cell lines intracranially in 2dpf zebrafish embryos to recapitulate an orthotopic model. Although this model is indeed clinically relevant, it requires sedation and precise positioning of embryonic zebrafish, making the transplantation process slow and technically challenging. Therefore, this pipeline cannot be readily implemented in high-throughput screens to profile novel candidate compounds. We thus ventured instead to inject the medulloblastoma cell lines into the 1k stage embryos using the validated pipeline of Van Bree et al. Injecting at the 1k stage requires no sedation nor precise orientation of the embryos and thus allows hundreds of embryos to be easily lined up in agarose moulds and transplanted quickly in comparison to the more technical orthotopic injection pipeline. Furthermore, it was showed by Van Bree et al and also in our study that transplanted medulloblastoma cells at the 1K stage do indeed show correct homing to the zebrafish brain making them a robust model to study the disease. Finally, our selection of the 1K stage transplantation was also due to its capacity to be implemented for high-throughput compound screens which can be of great interest both to the scientific community and the clinic due to its rapid timing.

3. Reviewer question:

Thank you for providing the additional data. Unfortunately it is still not clear if the treatment is reducing cell proliferation since cleaved PARP allows the assessment of apoptosis and there is no quantification of proliferation in the images provided in figure S. 2E. Could the authors quantify the proliferation in controls vs treatment conditions and show if there is a difference or not?

Authors reply: Thank you for pointing this out. We agree with this comment. Therefore, we have performed quantification of figure S. 2E, showing Ki67, cleaved Caspase 3 and cleaved PARP. This data is added as figure S. 2F (see also figure below). Our results show downregulation of KI67 after 24h treatment with BRQ IC50 in the D458 cells. In addition, there was significant upregulation of both cleaved Caspase 3 and cleaved PARP in D458 cells after 24h treatment with BRQ IC50 that was also confirmed at 72h treatment compared to control. Similar trends were noted in the case of D425 cells with significant upregulation of cleaved Caspase 3 after 72h treatment with BRQ and cleaved PARP after 24h treatment.

Round 3

Reviewer 3 Report

Comments and Suggestions for Authors